# Information, Beliefs, and Gender Stereotypes: Analysis of Socio-Cognitive Factors Influencing Healthcare for Intersex People

**DOI:** 10.3390/healthcare13222949

**Published:** 2025-11-17

**Authors:** Carla Palomino-Suárez, Marta Evelia Aparicio García

**Affiliations:** 1Faculty of Psicology, Somosaguas Campus, Complutense University of Madrid, 28223 Madrid, Spain; carpal04@ucm.es; 2Faculty of Medicine, Department of Psychology, CEU San Pablo University, Boadilla del Monte, 28668 Madrid, Spain; 3Instituto de Investigaciones Feministas (INSTIFEM), Complutense University of Madrid, Pabellón de Gobierno, 28040 Madrid, Spain

**Keywords:** intersex healthcare, medical education, gender stereotypes, socio-cognitive factors, human rights, professional attitudes

## Abstract

Intersex people continue to face barriers in healthcare. Despite notable ethical and legal advances, the role of socio-cognitive factors influencing clinical decision-making remains insufficiently understood. Critical perspectives call for revising the epistemological and normative foundations of medical practice, as clinical judgments may still be shaped by professionals’ beliefs and limited access to accurate information. **Objective:** This study examined how levels of knowledge, beliefs about gender determinism, and adherence to gender roles influence healthcare professionals’ attitudes toward intersex people. **Methods:** A total of 210 healthcare professionals from Spain participated in a cross-sectional survey. Participants completed the Intersex Knowledge Questionnaire, the short version of the Bem Sex-Role Inventory, and the Gender Determinism Scale. Data were analyzed using χ^2^ tests, one-way ANOVA, and *t*-tests. **Results:** Higher levels of knowledge (conceptual, procedural, and legislative) were associated with more affirmative and non-normative attitudes toward intersex healthcare. Neither gender determinism nor adherence to traditional gender roles was associated with professionals’ attitudes. Participants with prior contact with intersex people demonstrated higher conceptual knowledge and lower support for corrective surgeries. Significant disciplinary differences were also found: physicians tended to display more corrective and ambivalent attitudes, whereas psychologists and social workers were more frequently aligned with affirmative and diversity-respectful perspectives. **Conclusions:** Intersex healthcare attitudes may be influenced by limited training opportunities and the low visibility of intersex topics in medical education. Knowledge appears to be an important factor associated with more affirmative professional attitudes. Future studies using larger samples are needed to confirm these associations and explore underlying causal mechanisms.

## 1. Introduction

Intersex variations represent a natural expression of human bodily diversity. Their existence places intersex people in a position of vulnerability within social and medical systems historically designed around a dichotomous model of sex. Within this framework, the present study is part of a broader effort to examine whether healthcare professionals’ beliefs, attitudes, and knowledge influence the clinical practices applied to intersex people, and whether such practices may be shaped, beyond strictly medical criteria, by conformity to or identification with gender stereotypes and normative gender roles [1].

To situate this inquiry, this study adopts a socio-cognitive perspective that integrates classical theories of attitude formation [2], the emotional underpinnings of social cognition [3], and the internalization of gender stereotypes in professional behavior [4]. In the healthcare context, limited or distorted knowledge can interact with individual beliefs and perceived social norms, shaping clinical attitudes and, ultimately, influencing decision-making processes. Applying this perspective to intersex care makes it possible to understand professional responses not as isolated opinions but as cognitive systems structured by information, perceived norms, and socially shared stereotypes.

### 1.1. Medical History and the Normative Construction of Intersex Bodies

Intersex variations refer to a range of biological conditions in which an individual is born with sex characteristics that do not fit typical definitions of male or female. These differences may occur at the chromosomal, gonadal, hormonal, or morphological level. The persistence of a binary sex system has proven insufficient to capture this diversity, generating historical processes of invisibilization, pathologization, and stigmatization of intersex bodies [4,5].

Historically, medicine has interpreted intersex variations through a corrective and normalization-oriented paradigm, seeking to align the body with binary expectations of sex and gender. Until the 1950s, intersex variations were primarily understood through a pathological perspective, and early taxonomies were based on morphological classification criteria aimed at defining male or female categories [5]. From the mid-twentieth century onward, the prevailing model was sustained by the belief that gender identity was malleable and could be molded through medical intervention and socialization [6]. This rationale legitimized early surgical and hormonal procedures, often performed without proper informed consent from the patient or their guardians [7].

Since then, this medical model has been widely criticized for its inability to represent the complexity of intersex experiences and for overlooking the serious physical and psychological consequences of interventions aimed at “correcting” bodily diversity [1,8,9]. These practices have been questioned not only for their long-term impact and psychological well-being [1,10], but also for being grounded more in sociocultural constructs than in scientific evidence [4,8,11]. The conceptualization of intersex embodiment as a “disorder” has delayed legal and social progress [12], reinforcing stigma and invisibility. From a socio-cognitive perspective, this historical framework illustrates how clinical knowledge, cultural beliefs about gender, and prevailing social norms interacted to sustain these practices. Thus, early interventions can be understood as behavioral expressions of internalized belief systems rather than as evidence-based actions.

### 1.2. From the Corrective Model to Epistemological Critique: Questioning the Dominant Clinical Paradigm

Medical practice has long operated under the assumption that gender assignment, reinforced through early surgical intervention, would promote psychosocial well-being in adulthood. However, this premise lacks solid empirical support [1,10]. Viewed through a socio-cognitive lens, this assumption exemplifies how cognitive schemata and conformity pressures shape professional reasoning. When confronted with ethical doubts, healthcare providers may reinterpret such interventions as medically necessary “corrections” to maintain internal cognitive consistency with dominant cultural norms.

Within this context, various theoretical perspectives [4,13,14], invite a critical reassessment of the epistemological foundations that have guided medical discourse and practice. These perspectives emphasize that concepts of “normality” are socially constructed through language and institutional discourse, while professionals’ internalized beliefs and norms can unconsciously influence clinical judgment and behavior. Integrating these approaches highlights how epistemic frameworks, rather than neutral scientific structures, have historically reinforced binary models and legitimized interventionist practices. This enduring pathologization has also hindered the development of a culture of acceptance and respect for bodily diversity, compromising informed consent and ethical decision-making in clinical settings. When healthcare professionals operate under deterministic beliefs or uncritical adherence to gender stereotypes, intersex bodies are often perceived as problems to be “fixed” rather than as legitimate human variations [15]. Such cognitive rigidity distorts both the evaluation of medical evidence and the perceived legitimacy of interventions.

### 1.3. Legal Progress and International Recognition of Intersex Rights

In response to this situation, several legal frameworks have been developed to prohibit medically unnecessary interventions and safeguard the physical integrity of intersex people [16]. Over the past decade, international and national bodies have progressively recognized these interventions as violations of human rights rather than legitimate medical practices. The United Nations has taken concrete steps to protect the rights of intersex people, condemning non-consensual surgeries and other invasive procedures performed on intersex infants and children. This institutional acknowledgment marks a paradigm shift from medical paternalism to a rights-based approach.

Malta became the first European country to ban such practices in April 2015, establishing a legal precedent later followed by others. Iceland, for example, prohibited intersex genital mutilation on 19 July 2021. In Spain, the Law 4/2023 for the Real and Effective Equality of Trans and LGBTI People [17], introduced a specific provision banning non-consensual genital modifications in intersex minors under the age of 12, except in cases where such interventions are medically necessary to protect the individual’s health. These laws aim to protect the rights of intersex individuals [12] and to prevent surgical procedures lacking medical justification.

However, the global legal landscape remains uneven: in many regions, such interventions are still unregulated or legally permissible. At the same time, intersex rights movements in Latin America and the Caribbean have gained increasing visibility, successfully positioning intersex concerns on both international and national policy agendas. This growing advocacy has expanded public awareness and strengthened the call to dismantle binary conceptions of “male” and “female” across legal, medical, and cultural domains.

### 1.4. Gaps in Medical Training

In recent years, new clinical guidelines have been established to prohibit non-essential surgical interventions. Yet, despite these advances, limited empirical research has examined how healthcare professionals’ beliefs, attitudes, and the presence, or absence, of intersex-related content in medical education influence clinical decision-making.

According to the socio-cognitive framework guiding this study, three key variables interact dynamically in shaping professional behavior:Knowledge (conceptual, procedural, and legislative) provides the cognitive foundation for ethical and evidence-based decisions.Beliefs and stereotypes operate as cognitive filters that interpret or distort information.Attitudes emerge as evaluative tendencies that predispose behavior, influencing whether professionals adopt corrective or affirmative practices.

This interaction, knowledge, beliefs, and attitudes, constitutes the theoretical core of the present research. Following Ajzen’s (1991) [2], attitudes toward intersex care are expected to be related with knowledge and beliefs about gender and determinism.

### 1.5. Toward a Critical Medicine: The Need to Transform the Care Model

The guiding hypothesis of this study arises from a critical premise: what is considered clinically appropriate may be deeply conditioned by personal beliefs and by cultural frameworks that equate professional legitimacy with adherence to a binary model of sex and gender [18,19]. Incorporating socio-cognitive theory clarifies that these norms do not act abstractly but through internalized cognitive structures, that guide perception, judgment, and decision-making.

Thus, transforming healthcare for intersex people requires a dual approach:(1)An epistemological critique, to question the normative discourses that sustain the pathologization of bodily diversity;(2)Cognitive restructuring through education and reflective practice, enabling professionals to recognize bias and adopt diversity-affirming approaches.

This research therefore seeks to contribute empirical evidence supporting the transition toward a more ethical, informed, and respectful clinical model, aligned with international bioethical standards and the human rights of intersex people.

## 2. Materials and Methods

### 2.1. Objective

The present study aimed to examine whether and how healthcare professionals’ knowledge, beliefs about gender determinism, and adherence to traditional gender roles are related to their attitudes and clinical orientations toward intersex people.

Rather than establishing predictive or causal models, the study adopted a cross-sectional, comparative approach to identify significant associations and group differences across key socio-cognitive dimensions.

Specifically, the research (1) analyzed the relationship between the level of knowledge about intersex variations, across conceptual, procedural, and legislative domains, and clinical attitudes, to determine whether greater professional literacy corresponds to more affirmative and non-normative orientations.

It also (2) examined the role of beliefs about gender determinism and (3) adherence to traditional masculine and feminine roles, to explore whether these beliefs are linked to greater support for corrective medical practices or to reduced consideration of patient autonomy.

Furthermore, the study (4) assessed the potential influence of prior contact with intersex people, hypothesizing that such contact may relate to higher knowledge, more flexible gender conceptions, and more inclusive attitudes.

Finally, (5) differences across professional groups and medical specialties were explored to identify disciplinary patterns in levels of knowledge and attitudes toward intersex healthcare.

Based on these objectives, the following hypotheses were formulated:

**H1.** *Greater knowledge about intersex variations will be associated with more respectful and non-normative clinical practices toward intersex people*.

**H2.** *Higher levels of gender determinism will be associated with more normative and interventionist practices*.

**H3.** *Stronger adherence to traditional gender roles (normative masculinity or femininity) will be associated with greater support for corrective medical practices*.

**H4.** *Adherence to traditional gender roles (normative masculinity or femininity) will be associated with a greater tendency to consider informed consent in minors unnecessary, thereby legitimizing early surgical interventions without the direct participation of the child*.

**H5.** *Previous contact with intersex people will be associated with higher levels of knowledge and more respectful attitudes*.

**H6.** *Differences in levels of knowledge and attitudes toward intersex people are expected according to healthcare professionals’ group and medical specialty*.

### 2.2. Participants

The initial sample consisted of 247 participants. After a quality control review, 37 cases were excluded for not meeting the inclusion criteria or for completing the survey in less than five minutes, which was considered insufficient for providing valid responses. inclusion criteria required participants to be currently working, or to have worked, in healthcare or social care sectors, or to be in the final two years of a health-related university degree. Exclusion criteria included not belonging to these professional areas or reporting less than two years of study.

The final sample comprised 210 healthcare and social care professionals. Most participants identified as women (77.1%), with a mean age of 39.91 years (SD = 13.36). Nearly half were physicians from various specialties, including surgery (1.9%), gynecology (7.1%), primary care (9.0%), endocrinology (2.8%), pediatrics (9.5)%, and urology (1.9%). Other professional groups included nurses (18.6%), psychologists (17.6%), social workers (3.3%), physiotherapists (1.9%), and advanced students in health-related fields (9.0%). More than 40% of the participants reported over 15 years of professional experience, indicating a highly experienced sample.

Regarding sexual orientation, 70.5% identified as heterosexual, and in terms of marital status, 69.1% reported being married or in a stable relationship.

Descriptive sociodemographic and professional data are summarized in Table 1.

### 2.3. Instruments

Three instruments were used to assess participants’ gender role identification, beliefs about gender determinism, and knowledge related to intersex variations.

Bem Sex-Role Inventory, Short Version (BSRI) [20].

This instrument assesses the degree to which individuals identify with characteristics traditionally associated with masculinity and femininity. It consists of 20 items: 10 reflecting stereotypically masculine traits and 10 reflecting stereotypically feminine traits.

Previous studies have reported satisfactory reliability (α > 0.70). In the present sample, internal consistency was α = 0.74 for the masculinity subscale and α = 0.76 for the femininity subscale, confirming acceptable reliability for descriptive and comparative analyses.

Escala de Determinismo de Género (GDS) [21].

This scale assesses the extent to which individuals believe that gender is a fixed and biologically determined category. The scale has demonstrated good psychometric properties (α = 0.84 in prior studies), and internal consistency in the present sample was α = 0.82.

The Intersex Knowledge Questionnaire (IKQ) [8].

This questionnaire evaluates healthcare professionals’ knowledge and awareness concerning intersex variations. It includes items addressing conceptual understanding, socio-health procedures and clinical practices, and legal–jurisprudential frameworks, as well as exploratory questions about professional attitudes toward current medical approaches.

As this instrument has not yet undergone formal psychometric validation, it was employed in this study as an exploratory measure intended to capture preliminary indicators among healthcare professionals.

### 2.4. Procedure

Data were collected through an online questionnaire created using the Microsoft Forms platform, provided by the Complutense University of Madrid. A non-probabilistic sampling strategy combining convenience and snowball procedures was employed to reach healthcare professionals from diverse disciplines. Initial recruitment was conducted through direct contact with healthcare practitioners and university faculty affiliated with the research team, who represented less than 10% of the final sample. These participants were invited to complete the survey and to disseminate it within their professional networks.

To improve disciplinary balance, a post hoc stratified adjustment was later applied to ensure proportional representation across four professional areas—medicine, nursing, psychology, and social work—while maintaining sufficient subgroup sizes for comparative analyses. Additional responses were collected at professional conferences held in Madrid during 2025, as well as in hospitals and primary care centers, allowing inclusion of a wide range of clinical and health-related specialties.

Before participation, all respondents were informed of the study’s objectives, procedures, and voluntary nature. An informed consent form was presented at the start of the questionnaire, specifying that participation was anonymous, that no identifying personal or institutional data would be collected, and that responses would be used solely for research purposes. Participants were reminded that they could withdraw at any time by closing the browser window without saving their responses.

Confidentiality and data protection were guaranteed in accordance with the Spanish Organic Law 3/2018 on the Protection of Personal Data and the Guarantee of Digital Rights. A contact email address was provided for those requesting additional information or clarification about the study.

The study was conducted in accordance with the principles of the Declaration of Helsinki and was approved by the Ethics Committee of the Complutense University of Madrid (protocol code CE_20250508_02_SOC).

### 2.5. Data Analysis

To ensure the adequacy of the data for inferential analyses, the reliability of the instruments was assessed using Cronbach’s alpha coefficient, and sample normality was examined.

To facilitate comparisons, several composite variables were constructed:1.Knowledge. Operationalized through three subdimensions:Basic Concepts (participants’ understanding of intersex variations).Socio-Health Procedures (knowledge about medical practices and institutional recommendations).Legislation (familiarity with national and international regulations concerning the protection of intersex people’s rights).2.Attitudes. Defined by items assessing professionals’ agreement with performing corrective surgeries in childhood and the perceived importance of informed consent involving intersex minors.This index differentiated between normative attitudes (focused on adapting the body to binary gender expectations) and non-normative attitudes (emphasizing bodily integrity and autonomy).3.Gender Normativity. Derived from masculinity and femininity scores on the Bem Sex-Role Inventory (BSRI, Short Version). Cutoff points were based on sample means and standard deviations:Masculinity: M = 33.76, SD = 5.35 → high > 39.11; low < 28.41.Femininity: M = 40.41, SD = 5.33 → high > 45.74; low < 35.08.

Participants were classified as low normativity (balanced or low gendered traits), moderate normativity (partial inclination toward one role), or high normativity (clear alignment with traditional masculine or feminine traits).

Descriptive statistics (frequencies, percentages, means, and standard deviations) were calculated to summarize the sociodemographic and professional characteristics of the sample.

To test group differences, inferential analyses were performed as follows:

Chi-square (χ^2^) tests were used to examine associations between categorical variables, including knowledge levels (conceptual, procedural, and legislative) and attitudinal orientations toward intersex healthcare (H1), adherence to gender roles and support for corrective interventions (H3–H4), and professional group or specialty (H6).

One-way ANOVA was applied to compare mean differences in gender determinism and attitudinal orientations across multiple groups (H2).

Independent-samples t-tests were used to assess differences between two groups, particularly the effect of prior contact with intersex people on knowledge, attitudes, and gender determinism (H5).

Effect sizes (Cramér’s V for χ^2^, η^2^ for ANOVA, and Cohen’s d for *t*-tests) and 95% confidence intervals were reported to improve interpretability.

All analyses were conducted using IBM SPSS Statistics, version 29.

## 3. Results

### 3.1. General Overview of Participants’ Knowledge and Attitudes

A descriptive analysis was conducted to provide an overview of participants’ familiarity, attitudes, and levels of knowledge regarding intersex people and healthcare practices.

Approximately 23.3% of participants reported prior contact with an intersex person, and a large majority (80.5%) recognized intersex people as a vulnerable group within the healthcare system.

Attitudes toward medical practices revealed clear polarization. Nearly one-third of participants (30.5%) disagreed with performing corrective surgeries on newborns, whereas 29.0% considered such interventions justifiable under certain circumstances.

Concerning informed consent, more than half of respondents (56.9%) indicated being uncertain whether ethical standards are consistently upheld in the clinical management of intersex patients.

Overall, conceptual and procedural knowledge levels were generally low to moderate, while legislative knowledge presented the lowest scores, with 71.3% of participants scoring within the low range.

Nevertheless, 89.0% of participants expressed the need for specific professional training on intersex healthcare, underscoring a significant gap in both medical education and institutional frameworks. A summary of these descriptive data is presented in Table 2.

### 3.2. Relationship Between the Level of Knowledge About Intersex People and Attitudes Towards Them

Group comparison analyses revealed significant differences in clinical attitudes according to participants’ level of conceptual knowledge, χ^2^(12, N = 208) = 61.25, *p* < 0.001, Cramér’s V = 0.31. Corrective attitudes were observed more frequently among participants with low (46.7%) or no knowledge (26.7%), whereas they were notably less common among those with adequate (3.3%) or excellent (10.0%) knowledge. Similarly, indifferent or uninformed attitudes predominated among participants with no (52.2%) or low (39.1%) knowledge, while they were almost absent in higher knowledge groups.

Ambivalent attitudes were more common among those with low (40.8%) or medium (29.6%) knowledge but decreased among participants with adequate (15.5%) and excellent (5.6%) levels. In contrast, affirmative or non-normative attitudes increased as knowledge improved, being most frequent among participants with adequate (23.0%) and excellent (13.1%) knowledge, compared with those reporting no (8.2%) or low (29.5%) knowledge.

The linear trend test was also significant, χ^2^(1) = 26.30, *p* < 0.001, indicating that higher levels of knowledge were associated with affirmative and non-normative attitudes.

Overall, as knowledge increased, attitudes shifted toward more respectful and inclusive positions (see Table 3).

Significant differences were found in attitudes toward intersex care based on levels of procedural knowledge, χ^2^(9, N *=* 208) = 108.91, *p* < 0.001, Cramér’s V *=* 0.42. Corrective attitudes were most frequent among participants with low (45.2%) or no procedural knowledge (38.7%) and were considerably less common among those with medium (6.5%) or excellent (9.7%) knowledge. Similarly, indifferent or uninformed attitudes predominated among those with no procedural knowledge (58.7%) but decreased notably at medium (23.9%) and excellent (10.9%) levels.

In contrast, ambivalent attitudes were more prevalent among participants with medium (38.0%) and excellent (43.7%) levels of procedural knowledge compared to those with none (4.2%) or low (14.1%) knowledge. Finally, affirmative or non-normative attitudes increased with higher levels of procedural knowledge, being most frequent among participants with excellent (53.3%) and medium (36.7%) knowledge, while nearly absent among those with none (1.7%) or low (8.3%) knowledge. The linear trend test was also significant, χ^2^(1) = 66.30, *p* < 0.001, indicating that higher procedural knowledge was associated with a greater likelihood of expressing affirmative and non-normative attitudes (see Table 4).

Finally, significant differences were also found in attitudes toward intersex care as a function of legislative knowledge, χ^2^(9, N = 209) = 28.21, *p* < 0.001, Cramér’s V = 0.21. Participants with no legislative knowledge were overrepresented among those who exhibited corrective attitudes (77.4%) and, especially, indifferent or uninformed attitudes (95.7%), as well as comprising the majority of those with ambivalent attitudes (69.0%).

In contrast, participants with excellent knowledge of legislation were more likely to endorse affirmative or non-normative attitudes (18.0%), compared with only 6.5% holding corrective and 2.2% holding indifferent positions. Those with low or medium levels of legislative knowledge occupied intermediate positions, accounting for 23.9% of ambivalent and 27.9% of affirmative/non-normative attitudes.

The linear trend test was also significant, χ^2^(1) = 15.50, *p* < 0.001, suggesting that higher levels of legislative knowledge were systematically associated with a greater likelihood of expressing affirmative and non-normative attitudes, and a lower likelihood of supporting corrective or indifferent positions (see Table 5).

These results support H1, suggesting that higher levels of knowledge about intersex topics, whether conceptual, procedural, or legislative, are associated with more respectful, affirmative, and non-normative clinical attitudes.

### 3.3. Relationship Between Gender Determinism and Traditional Gender Roles with Attitudes Toward Intersex People, Justification of Corrective Surgeries, and Perceptions of Valid Informed Consent

No significant differences were found in mean gender determinism scores across the four attitudinal categories. In addition, an analysis was conducted to explore whether considering corrective surgeries as appropriate (yes vs. no, excluding intermediate response options) was associated with gender determinism. Results indicated no significant differences in determinism scores between the two groups, *t*(88) = 0.69, *p* = 0.245, Cohen’s *d* = 0.15, 95% CI [−0.60, 1.23]. Both analyses suggest that gender determinism did not differentiate between corrective, indifferent, ambivalent, or affirmative/non-normative attitudes, nor was it related to the acceptance or rejection of corrective surgeries among participants.

Regarding the analysis of adherence to traditional gender roles and its relationship with greater support for corrective medical practices or with the belief that informed consent is properly obtained in surgeries performed on minors (addressed by Objectives 3 and 4), results showed no significant associations either between adherence to traditional gender roles and approval of corrective surgeries or between gender-role adherence and the perception that informed consent requirements are fulfilled in such interventions, χ^2^(18, N = 209) = 15.28, *p* = 0.643; χ^2^(18, N = 209) = 26.30, *p* = 0.093.

The results do not support H2–H4, as neither gender determinism nor adherence to traditional gender roles showed significant associations with corrective attitudes or perceptions of informed consent in minors.

### 3.4. Prior Contact with Intersex People and Its Influence on Healthcare Professionals’ Knowledge and Attitudes

Prior contact with intersex people was associated with higher conceptual knowledge, lower support for corrective interventions, and a more flexible understanding of gender identity construction. Participants who reported previous contact obtained higher scores in basic conceptual knowledge (M = 1.82, SD = 1.20) compared to those without such contact (M = 1.49, SD = 1.03), *t*(148) = 1.75, *p* = 0.041. Similarly, participants with contact expressed lower support for corrective surgeries (M = 2.78, SD = 0.80) than those without contact (M = 2.98, SD = 0.65), *t*(149) = −1.69, *p* = 0.047.

Moreover, participants with previous contact were less likely to endorse the belief that gender identity is fixed (M = 1.55, SD = 0.79) than those without contact (M = 1.83, SD = 0.89), *t*(149) = −1.89, *p* = 0.030. In contrast, no significant differences were observed in legislative knowledge between participants with (M = 0.51, SD = 0.71) and without prior contact (M = 0.43, SD = 0.71), *t*(149) = 0.63, *p* = 0.262 (see Table 6).

These results support H5, showing that prior contact with intersex people is related to higher conceptual knowledge and more respectful, non-normative attitudes.

### 3.5. Differences in Knowledge Toward Intersex People Among Professional Groups

Comparative analyses revealed significant differences in conceptual knowledge across professional groups, χ^2^(20, N = 209) = 34.68, *p* = 0.016, Cramér’s V *=* 0.21. Physicians displayed the highest levels of conceptual understanding, representing 42.3% of participants with adequate knowledge and 81.3% of those with excellent knowledge.

In contrast, nurses and students were more frequently concentrated in the lowest knowledge categories, while psychologists and social workers tended to cluster at intermediate levels. Physiotherapists appeared only sporadically across the different knowledge categories, reflecting their limited representation in the sample (see Table 7).

No statistically significant associations were found between professional group and other knowledge domains. Specifically, differences were non-significant observed for procedural knowledge, χ^2^(16, N = 208) = 18.37, *p* = 0.252, V = 0.17, and for legislative knowledge, χ^2^(12, N = 209) = 18.56, *p* = 0.102, V = 0.19.

These results support H6, suggesting that while conceptual understanding of intersex-related topics varies moderately among professional disciplines, procedural and legislative knowledge remain uniformly limited across groups.

### 3.6. Differences in Attitudes Toward Intersex People Among Professional Groups

The chi-square analysis revealed a significant association between professional group and attitudes toward intersex people, χ^2^(15, N = 209) = 32.00, *p* = 0.024, Cramér’s V = 0.21. Physicians were overrepresented in corrective and ambivalent attitudes (67.7% and 50.7%, respectively), indicating greater support for interventionist perspectives. Psychologists, in contrast, were disproportionately concentrated in the affirmative and non-normative group (34.4%), reflecting a greater alignment with diversity-affirming approaches.

Nurses showed a higher frequency of indifferent or uninformed responses (28.3%), while social workers clustered mainly in the affirmative/non-normative category (6.6%). Students exhibited a more balanced distribution, with a notable presence in ambivalent positions (12.7%).

Overall, the results support H6, indicating that attitudinal orientations toward intersex healthcare vary across professions, with medical disciplines showing greater adherence to corrective paradigms, while psychology and social work emphasize more inclusive and diversity-oriented perspectives (see Table 8).

The results support H6. Knowledge and attitudes toward intersex care vary by profession: physicians showed higher knowledge and more corrective views, while psychologists and social workers held more affirmative attitudes.

### 3.7. Differences in Knowledge and Attitudes Toward Intersex Care Across Medical Specialties

To further explore these findings, the sample was segmented by medical specialty among those directly involved in the attention of intersex people, forming six groups: primary care, surgery, gynecology, endocrinology, urology, and pediatrics. Chi-square analyses were conducted to examine possible differences in levels of knowledge and attitudes across specialties. No statistically significant differences were found for conceptual knowledge, χ^2^(20, N = 70) = 18.02, *p* = 0.550; procedural knowledge, χ^2^(15, N = 69) = 18.16, *p* = 0.207; legislative knowledge, χ^2^(10, N = 70) = 8.16, *p* = 0.620; or attitudes toward intersex care, χ^2^(15, N = 70) = 22.09, *p* = 0.071.

Although these associations did not reach statistical significance, the distribution of responses revealed distinct professional patterns worthy of attention. Surgeons and urologists exhibited the most polarized profiles: surgeons were divided between corrective and affirmative positions, whereas urologists concentrated mainly in corrective and indifferent attitudes, with almost no representation in non-normative positions. Gynecologists also displayed a polarized pattern, with approximately one-quarter supporting corrective practices and another quarter endorsing non-normative approaches.

Pediatricians emerged as the group showing the highest support for corrective attitudes (33.3%) and, simultaneously, the highest proportion of indifferent responses (50.0%), reflecting the persistent tension in pediatric practice between traditional interventionist models and emerging critical perspectives. In contrast, primary care and endocrinology demonstrated more critical and reflective profiles, characterized by higher proportions of ambivalent and non-normative attitudes and minimal support for corrective interventions.

Overall, although differences did not reach statistical significance, the observed distribution suggest disciplinary tendencies: pediatrics and surgical fields remain closer to the interventionist legacy, whereas primary care and endocrinology exhibit greater openness to diversity-affirming perspectives (see Table 9).

## 4. Discussion

The present findings underscore persistent barriers in healthcare for intersex people and revealed critical deficits in professional knowledge and training. Although many healthcare professionals reported inclusive attitudes, most acknowledged insufficient preparation to provide competent and ethically informed care. Within this context, limited knowledge appears to translate into professional uncertainty, which may contribute to the persistence of traditional practices whose clinical validity remains questionable.

The results indicate that lower levels of conceptual, procedural, and legislative knowledge are associated with more corrective or indifferent attitudes, whereas higher knowledge levels correspond to more affirmative and diversity-respectful positions, confirming hypothesis 1. However, these associations should be interpreted descriptively rather than causally, given the cross-sectional and non-predictive design. It is plausible that professionals with more inclusive attitudes actively seek further training, suggesting a bidirectional relationship. This pattern aligns with previous research highlighting that insufficient understanding of bodily and sexual diversity fosters the reproduction of stereotypes and interventionist practices [9,22,23].

Procedural knowledge emerged as a particularly relevant dimension. Professionals who possess clear protocols and experience with inclusive clinical practices tend to exhibit greater confidence and more respectful attitudes [24]. In contrast, a lack of procedural knowledge may lead practitioners to rely on culturally ingrained heuristics or binary biomedical models, perpetuating corrective approaches derived from outdated paradigms [25]. The absence of established guidelines not only heightens professional uncertainty but also increases the likelihood of medically unnecessary interventions, with long-term consequences that often result in patient mistrust and avoidance of future care [26].

A striking finding concerns the widespread deficit of legislative knowledge, observed across all professional profiles. This gap limits awareness of the human rights frameworks that should orient medical decisions. International recommendations consistently emphasize the principles of bodily integrity and informed consent as the foundation for ethical intersex care [27,28,29]. Without integrating these principles into training, professionals may inadvertently legitimize practices inconsistent with current ethical standards. These results support the urgent need to incorporate ethical and legal literacy into medical education, ensuring alignment between clinical practice and international human rights directives.

However, the relationship between knowledge and attitudes is not linear. At intermediate knowledge levels, ambivalent responses emerged, suggesting a transitional stages where professionals begin to question normative assumptions yet lack the conceptual and procedural tools for consistent affirmative practice. This pattern is consistent with previous studies showing that partial awareness, when not supported by structured education, can lead to inconsistent decision-making [9,24,30]. Comprehensive and sustained training is therefore essential to move beyond superficial sensitivity toward genuine competence.

Overall, the results support prior evidence linking educational deficits with the persistence of normative healthcare models. The absence of systematic education on sexual and gender diversity perpetuates the medicalization of intersex variations [22]. Qualitative studies further indicate that these educational gaps are associated with invisibility, non-consensual treatments, and violations of fundamental rights [9,31]. The present findings extend this body of evidence but should be read as indicative associations, not causal effects.

A notable result is that neither gender determinism nor adherence to traditional masculine or feminine roles was associated with healthcare professionals’ attitudes, providing no support for Hypotheses 2–4. This suggests that personal beliefs may not currently be decisive factors shaping clinical attitudes, contrasting with classical research that linked essentialist beliefs to prejudice toward non-normative identities [32,33,34,35]. It also challenges prior hypotheses suggesting that a fixation on binary categories underlies the historical tendency to recommend “normalizing” surgeries on intersex bodies [25]. Instead, other socio-cognitive variables—such as specific clinical knowledge, institutional guidelines, and experience—may exert stronger influence. Even professionals with traditional gender beliefs may adjust their practices when provided with updated, rights-based clinical frameworks.

Considering the broader cultural and institutional context of Spain, recent data indicate that society as a whole no longer adheres to traditional gender roles as strongly as in the past, although tensions and resurgences persist among certain groups, particularly younger men [36]. In this context, gender stereotypes no longer appear to determine professional behavior automatically; rather, decisions seem increasingly guided by biomedical training and institutional standards [37].

The finding that gender beliefs did not relate with attitudes toward corrective surgeries may also be interpreted as a correlational indication of a broader cultural and generational shift within medicine. Although the gender binary remains the dominant social framework, early interventions on intersex minors are increasingly questioned from ethical and human rights perspectives [31,37].

Consistent with Hypothesis 5, prior contact with intersex people was associated with higher levels of conceptual knowledge, lower support for corrective interventions, and lower endorsement of gender determinism. Nonetheless, these relationships should not be interpreted as unidirectional. These findings support the idea that direct interaction may contribute to reducing prejudice and fostering more respectful attitudes. In line with intergroup contact theory [38]. Exposure combined with institutional support may reduce social distance and encourage recognition of diversity as a natural human variation [39,40,41,42]. Quinn (2015) [41] and colleagues have further examined knowledge gaps and attitudinal barriers among health professionals toward LGBTQ+ patients, emphasizing the need for educational interventions to reduce institutional prejudice [41,42]. However, such contact did not improve legislative knowledge, underscoring that interpersonal experience alone cannot substitute for formal education on legal and ethical rights. This structural deficit persists across healthcare curricula and requires institutional reform [25,37].

In support of Hypothesis 6, the results of this study reveal significant professional differences in both knowledge and attitudes. In terms of knowledge, physicians occupied the highest positions in conceptual understanding, while nurses and students tended to cluster at the lowest levels, and psychologists and social workers were more frequently positioned at intermediate levels. This pattern suggests that medical education provides a stronger conceptual foundation. Nevertheless, these disciplinary contrasts should be interpreted descriptively.

However, it is particularly noteworthy that no differences were observed between professional groups in procedural or legislative knowledge. This finding indicates that educational gaps are transversal, affecting all professional profiles similarly. This result is particularly concerning for physicians, who hold primary responsibility for clinical decision-making regarding intersex people and therefore require a comprehensive command of clinical protocols and the legal frameworks governing medical practice. The absence of differences across professions suggests that even those with greater conceptual knowledge often lack the applied tools and legal literacy necessary to align decisions with human rights standards and international recommendations.

At the attitudinal level, however, clearer distinctions emerged. Physicians were disproportionately concentrated in corrective and ambivalent positions, reflecting greater adherence to interventionist and “normalizing” models. This pattern aligns with prior research documenting how the biomedical tradition—centered on bodily correction—has historically shaped clinical practice through a deeply entrenched interventionist bias [25,31]. In contrast, psychologists predominantly expressed affirmative and non-normative attitudes, consistent with training emphasizing psychosocial dimensions and personal autonomy, which enhances ethical sensitivity toward bodily diversity [37].

Nurses, by contrast, showed higher prevalence of indifferent or uninformed responses, a trend consistent with evidence highlighting the lack of specific education on sexual and bodily diversity in nursing curricula [24]. Social workers tended to align with affirmative perspectives, consistent with their historical role in rights advocacy and community-based care. Students displayed heterogeneous profiles, with many occupying ambivalent positions, a likely reflection of a transitional stage in their professional development, where normative practices are questioned but not yet replaced by fully affirmative frameworks.

The analysis of medical specialties revealed additional professional patterns. Surgical fields—particularly surgery and urology—remained closer to corrective and indifferent attitudes, consistent with literature identifying surgical practice as a central locus of the interventionist legacy in intersex healthcare [25,37].

Pediatrics emerged as especially significant. Pediatricians simultaneously showed high support for corrective interventions and a notable proportion of indifferent responses, reflecting persistent tensions within the specialty. As the first point of contact for many intersex cases, pediatricians often face institutional and familial pressure, which has historically led to hasty and poorly informed decisions [5]. The persistence of indifferent responses further suggests ongoing deficits in specific training, leaving many pediatricians without adequate tools to adopt a critical stance [43].

In contrast, primary care and endocrinology exhibited more reflective and open profiles, characterized by greater proportions of ambivalent or non-normative attitudes and minimal support for corrective procedures. This trend may be explained by the type of clinical relationship these professionals maintain—longitudinal, dialogic, and oriented toward accompaniment rather than intervention. Continuous engagement with patients and families in non-surgical contexts may facilitate greater questioning of traditional medical norms [44].

Gynecology displayed a transitional pattern, with roughly equal representation of corrective and affirmative attitudes. This division reflects a discipline undergoing change: historically implicated in normalizing interventions, yet increasingly engaged in debates on autonomy, consent, and sexual health [5]. The coexistence of these contrasting orientations may be interpreted as a sign of ongoing internal transformation, though not yet fully consolidated.

## 5. Limitations

This study has several limitations. Although the sample size was adequate for the planned statistical analyses and included professionals from multiple healthcare disciplines, recruitment relied on voluntary participation through professional networks and online distribution. Consequently, the sample could not be considered fully representative. The use of a non-probabilistic and self-selected sample may have introduced selection bias, as those with greater sensitivity or prior interest in gender and diversity issues might have been more likely to participate. Conversely, it is not possible to determine whether non-respondents declined due to lack of knowledge about intersex topics or simply a lack of interest in the topic, which could have affected the diversity of perspectives included.

The non-probabilistic nature of the sample design limits the representativeness of the findings. Although post hoc stratification improved disciplinary balance, selection bias cannot be fully ruled out. The present results should therefore be interpreted as exploratory and descriptive rather than generalizable to all healthcare professionals in Spain. Future studies should employ quota-based sampling to enhance external validity and allow for more robust cross-disciplinary comparisons.

Another limitation concerns the scarcity of validated instruments specifically designed to assess intersex-related knowledge and attitudes. The Intersex Knowledge Questionnaire used in this study has not yet undergone formal psychometric validation; however, it was developed based on a comprehensive review of international guidelines, academic literature, and ethical-legal recommendations related to intersex healthcare. Its items were reviewed to ensure content relevance and conceptual clarity. Despite its methodological rigor, the absence of internationally standardized and validated tools in this field limits the comparability of results across studies. Future research should therefore prioritize the development and cross-cultural validation of psychometrically robust instruments capable of capturing conceptual, procedural, and legislative dimensions of intersex healthcare.

Finally, although the inclusion of different medical specialties strengthens the study’s ecological validity, contextual variables such as institutional culture, local clinical guidelines, or exposure to intersex people were not systematically controlled. These unmeasured factors may moderate the observed relationships and should be incorporated into future research.

## 6. Conclusions

This study provides descriptive evidence that clarifies the relationships between knowledge, beliefs, and professional attitudes in the context of intersex healthcare, an area still marked by limited research and persistent normative and educational tensions [25,31,37].

The findings show that conceptual, procedural, and legislative knowledge are variables associated with professional attitudes. Higher literacy in these areas coincided with lower support for corrective interventions and with greater openness to affirmative and diversity-affirming approaches. Nevertheless, the results reveal a cross-disciplinary deficit in training, particularly in procedural and legislative knowledge, even among professionals whose roles require this expertise to ensure safe and rights-based care.

While previous contact with intersex people was linked to higher conceptual understanding and less support for “normalizing” surgeries, this experiential factor did not appear to be related to greater awareness of legal or ethical aspects. This pattern suggests that experiential learning alone may be insufficient and should be complemented by formal education in applied ethics and legal frameworks [37,45].

Overall, the results indicate that professional attitudes may be shaped more by the presence or absence of structured training and institutional frameworks than by individual gender beliefs. In contexts where educational and regulatory references are limited, gender-binary assumptions may still influence clinical reasoning. Conversely, stronger and more comprehensive training appears to coincide with affirmative and ethically aligned attitudes [9].

Although disciplinary differences did not reach statistical significance across all analyses, the observed tendencies provide useful interpretative insights. Surgical and pediatric specialties showed patterns closer to interventionist traditions, whereas primary care and endocrinology displayed a greater tendency toward non-normative and diversity-affirming positions. Psychosocial fields appeared more aligned with rights-based approaches. These results can be interpreted as reflecting disciplinary socialization processes that are open to modification through targeted educational and institutional policies [37].

In summary, the main area for intervention lies not in individual willingness but in the structural need to strengthen training and regulatory frameworks. The evidence supports the development of curricular programs that integrate conceptual, procedural, and legislative components, together with educational experiences involving direct interaction with intersex people to promote affirmative and respectful clinical practices. Improving education, regulation, and institutional protocols thus emerges as a realistic and effective pathway for aligning clinical work with ethical and human-rights standards [22,24,41].

Future research should prioritize the development and validation of standardized instruments, especially a psychometrically tested version of the Intersex Knowledge Questionnaire, since the current tool was exploratory. Validated measures would allow reliable comparisons across contexts and over time. Further studies should also examine the effects of educational initiatives, explore emotional and cognitive components of professional attitudes, and include intersex people’s perspectives to advance an inclusive, rights-based model of healthcare.

## Figures and Tables

**Table 1 healthcare-13-02949-t001:** Sociodemographic and Professional Characteristics of the Sample (N = 210).

Variable	Categories	n	%
Sex	Woman	162	77.1
	Man	44	21.0
	Prefer not to answer	4	1.9
Profession	Health/social sciences student	19	9.0
	Nurse	39	18.6
	Psychologist	37	17.6
	Social worker	7	3.3
	Physiotherapist	4	1.9
	Physician (various specialties)	104	49.5
Professional experience	<2 years	13	6.2
	2–5 years	23	11.0
	5–10 years	33	15.7
	10–15 years	24	11.4
	>15 years	86	41.0
	Student	23	11.0
	None of the above	7	3.3
Sexual orientation	Heterosexual	148	70.5
	Homosexual	24	11.4
	Bisexual	31	14.8
	Other	2	1.0
	Prefer not to answer	5	2.4
Marital status	Single	45	21.4
	Partnered	59	28.1
	Married/civil union	86	41.0
	Separated/divorced	17	8.1
	Widowed	1	0.5
	Prefer not to answer	2	1.0

**Table 2 healthcare-13-02949-t002:** Attitudes, Knowledge, and Perceptions Regarding Intersex Care (N = 210).

Variable	Categories	n	%
Contact with intersex people	Yes	49	23.3
	No	161	76.7
Perception of vulnerability	Considered vulnerable	169	80.5
Corrective surgeries in newborns	Disagree	64	30.5
	Agree	26	12.4
	Sometimes justified	61	29.0
	Do not know	58	27.6
Informed consent compliance	Yes	10	4.8
	No	74	35.4
	Sometimes	6	2.9
	Do not know	120	56.9
Knowledge: Concepts	Low	124	58.9
	Medium	44	21.1
	High	42	20.0
Knowledge: Procedures	Low	76	36.1
	Medium	63	29.8
	High	71	34.1
Knowledge: Legislation	Low	150	71.3
	Medium	40	19.1
	High	19	9.1
Need for training	Yes	187	89.0
	No	23	11.0

**Table 3 healthcare-13-02949-t003:** Distribution of Attitudes toward Intersex Care by Levels of Knowledge of concepts.

			Attitudes				Total
			Corrective	Indifferent/Uninformed	Ambivalent	Affirmative	
Conceptual Knowledge	None	N	8	24	6	5	43
		%	26.7%	52.2%	8.5%	8.2%	20.7%
	Low	N	14	18	29	18	79
		%	46.7%	39.1%	40.8%	29.5%	38.0%
	Medium	N	4	3	21	16	44
		%	13.3%	6.5%	29.6%	26.2%	21.2%
	Adequate	N	1	0	11	14	26
		%	3.3%	0.0%	15.5%	23.0%	12.5%
	Excellent	N	3	1	4	8	16
		%	10%	2.2%	5.6%	13.1%	7.7%
Total		N	30	46	71	61	208
		%	100%	100%	100%	100%	100%

**Table 4 healthcare-13-02949-t004:** Distribution of Attitudes toward Intersex Care by Levels of knowledge of sociosanitary procedures.

			Attitudes				Total
			Corrective	Indifferent/Uninformed	Ambivalent	Affirmative	
Procedural Knowledge	None	N	12	27	3	1	43
		%	38.7%	58.7%	4.2%	1.7%	20.7%
	Low	N	14	3	10	5	32
		%	45.2%	6.5%	14.1%	8.3%	15.4%
	Medium	N	2	11	27	22	62
		%	6.5%	23.9%	38.0%	36.7%	29.8%
	Excellent	N	3	5	31	32	71
		%	9.7%	10.9%	43.7%	53.3%	34.1%
Total		N	31	46	71	60	208
		%	100%	100%	100%	100%	100%

**Table 5 healthcare-13-02949-t005:** Distribution of Attitudes toward Intersex Care by Levels of Legislative Knowledge.

			Attitudes				Total
			Corrective	Indifferent/Uninformed	Ambivalent	Affirmative	
Legislation Knowledge	None	N	24	44	49	32	149
		%	77.4%	95.7%	69.0%	52.5%	71.3%
	Low/medium	N	5	1	17	17	40
		%	16.1%	2.2%	23.9%	27.9%	19.1%
	Excellent	N	2	1	5	11	19
		%	6.5%	2.2%	7.0%	18.0%	9.1%
Total		N	31	46	71	61	209
		%	100%	100%	100%	100%	100%

**Table 6 healthcare-13-02949-t006:** Independent-Samples t-tests of Knowledge, corrective interventions and gender determinism by Prior Contact with Intersex People.

Variable	Contact Yes M (SD)	Contact No M (SD)	*t*	df	*p*
Concepts Knowledge	1.82 (1.20)	1.49 (1.03)	1.75	148	0.041
Corrective surgeries	2.78 (0.80)	2.98 (0.65)	−1.69	149	0.047
Belief in fixed gender identity	1.55 (0.79)	1.83 (0.89)	−1.89	149	0.030
Legislation Knowledge	0.51 (0.71)	0.43 (0.71)	0.63	149	0.26

**Table 7 healthcare-13-02949-t007:** Distribution of Concepts Knowledge by Profession.

			Concepts Knowledge					Total
			None	Low	Medium	Adequate	Excellent	
Profession	Students	N	5	7	0	6	1	19
		%	11.4%	8.9%	0.0%	23.1%	6.3%	9.1%
	Nurses	N	12	17	8	0	1	38
		%	27.3%	21.5%	18.2%	0.0%	6.3%	18.2%
	Physicians	N	17	38	25	11	13	104
		%	38.6%	48.1%	56.8%	42.3%	81.3%	49.8%
	Psychologists	N	7	13	11	5	1	37
		%	15.9%	16.5%	25.0%	19.2%	6.3%	17.7%
	Social workers	N	2	3	0	2	0	7
		%	4.5%	3.8%	0.0%	7.7%	0.0%	3.3%
	Physiotherapists	N	1	1	0	2	0	4
		%	2.3%	1.3%	0.0%	7.7%	0.0%	1.9%
Total		N	44	79	44	26	16	209
		%	100%	100%	100%	100%	100%	100%

**Table 8 healthcare-13-02949-t008:** Distribution of Attitudes by Profession.

			Attitudes				Total
			Corrective	Indifferent/Uninformed	Ambivalent	Affirmative	
Profession	Students	N	3	4	9	3	19
		%	9.7%	8.7%	12.7%	4.9%	9.1%
	Nurses	N	5	13	13	8	39
		%	16.1%	28.3%	18.3%	13.1%	18.7%
	Physicians	N	21	22	36	24	103
		%	67.7%	47.8%	50.7%	39.3%	49.3%
	Psychologists	N	1	5	10	21	37
		%	3.2%	10.9%	14.1%	34.4%	17.7%
	Social workers	N	0	1	2	4	7
		%	0.0%	2.2%	2.8%	6.6%	3.3%
	Physiotherapists	N	1	1	1	1	4
		%	3.2%	2.2%	1.4%	1.6%	1.9%
Total		N	31	46	71	61	209
		%	100%	100%	100%	100%	100%

**Table 9 healthcare-13-02949-t009:** Distribution of Attitudes toward Intersex Care by Medical Specialties.

			Attitudes				Total
			Corrective	Indifferent/Uninformed	Ambivalent	Affirmative	
Medical Specialties	Surgery	N	2	0	1	1	4
		%	13.3%	0.0%	4.0%	6.3%	5.7%
	Gynecology	N	4	1	6	4	15
		%	26.7%	7.1%	24.0%	25.0%	21.4%
	Primary care	N	2	3	10	4	19
		%	13.3%	21.4%	40.0%	25.0%	27.1%
	Endocrinology	N	1	0	5	2	8
		%	6.7%	0.0%	20.0%	12.5%	11.4%
	Pediatrics	N	5	7	3	5	20
		%	33.3%	50.0%	12.0%	31.3%	28.6%
	Urology	N	1	3	0	0	4
		%	6.7%	21.4%	0.0%	0.0%	5.7%
Total		N	15	14	25	16	70
		%	100.0%	100.0%	100.0%	100.0%	100.0%

## Data Availability

The dataset used and analyzed in this study is available from the corresponding author. The data are not publicly available in accordance with the ethical approval, which specifies that the dataset must be stored securely, in compliance with institutional data protection procedures.

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
