# Peer review of "Information, Beliefs, and Gender Stereotypes: Analysis of Socio-Cognitive Factors Influencing Healthcare for Intersex People"

_healthcare, 2025, doi:10.3390/healthcare13222949_

Round 1

Reviewer 1 Report

Comments and Suggestions for Authors

This paper takes on an important and often overlooked topic: how beliefs and knowledge shape healthcare professionals’ attitudes toward intersex people. It’s thoughtful, clearly written, and solidly within the journal’s scope on health, ethics, and practice. That said, some conceptual, methodological, and structural issues make the findings harder to trust and interpret as they stand. With a substantial revision - tightening the framework, clarifying methods, and streamlining the structure - the manuscript could become a strong contribution.

  1. The introduction functions as a mini-review, repeating points on pathologization/medicalization and the history of intersex interventions. While the theoretical framing (Butler, Fausto-Sterling, Karkazis) is valuable, its length is disproportionate to the empirical section.
  2. The paper invokes "sociocognitive factors" but doesn't manage to anchor them in a clear theoretical model. Concepts like attitudes, knowledge, and beliefs are treated descriptively rather than linked within a coherent framework. To strengthen the work, articulate and operationalize a specific theory that explicitly connects cognitive variables to attitudes and behaviors.
  3. The study relies on snowball and convenience sampling followed by post hoc "stratification", a methodologically inconsistent approach that undermines representativeness. Although selection bias is acknowledged, its potential impact is not quantified. The spanish sample is then used to make crossdisciplinary claims, while the distribution across medical specialties remains unclear. To address these issues, specify recruitment channels and inclusion/exclusion criteria, clarify specialty representation, and justify the adequacy of the 210 participants for testing six hypotheses.
  4. The Intersex Knowledge Questionnaire is used without evidence of validation, and no psychometric properties are reported (e.g., factor structure, subscale reliability). "Attitudes" are operationalized with a small set of corrective vs. affirmative items that lack psychometric grounding. Category thresholds ("excellent", "adequate", etc.) appear to derive from mean SD, which can distort classification. Please report internal consistency (α, ω), conduct dimensionality analyses (EFA/CFA), and clarify whether any scales were adapted/translated and pretested.
  5. The analysis addresses complex, multivariate questions, but reliance on chi-square tests and t-tests limits interpretability. Clarity would improve by using multivariable models (e.g., multiple regression or SEM) that adjust for key covariates such as age, profession, gender, and prior contact. Effect sizes and confidence intervals should be reported consistently, and the use of one-tailed p-values should be justified or avoided. These steps would yield a stronger, more nuanced picture of the results.
  6. The crosssectional design does not support causal inference, yet several conclusions imply directional effects (e.g., "knowledge drives attitudes"). The discussion should also consider reverse causality, whereby professionals with more inclusive attitudes may be more likely to seek training. Rephrase claims to reflect correlational rather than causal relationships.
  7. Terminology is inconsistent: "intersexuality" and "intersex variations" are used interchangeably. Current practice favors intersex or variations of sex characteristics (VSC). Please standardize throughout and explain your usage at first mention. Ethical approval is reported, but the procedures for online recruitment - especially anonymity, data protection, and informed consent - need clearer description.
  8. The conclusions overstate the study’s implications given its methodological limitations. In particular, the claims that "Intersex healthcare attitudes are primarily driven by deficits 28 in training and the invisibility of intersex issues in medical education" (Abstract) and "These findings confirm that clinical attitudes are not explained solely by gender ste-730 reotypes but are primarily shaped by the availability of specialized training, institutional 731 organization, and the presence, or absence, of clear normative frameworks" are intriguing but not supported by sufficiently robust analyses. Please temper the language and align conclusions more closely with the descriptive evidence presented.
  9. Tables 3–9 are overly detailed and repetitive. Several could be moved to Supplementary Material.
  10. Reference list shows inconsistencies (missing DOIs, uneven citation formatting).
  11. Abstract should include sample size and main statistical results.
  12. Some citations appear misformatted or missing reference details. See: Butler, J. El género en disputa: el feminismo y la subversión de la identidad; Paidós estudio; [1a ed., 12a reimpr.].; Paidós: Barcelona, 803 2020; ISBN 9788449320309.
Comments on the Quality of English Language

English is generally good but should undergo professional language editing to improve conciseness and flow.

Author Response

Thank you very much for taking the time to review this manuscript. We sincerely appreciate your feedback and suggestions. Your comments have been extremely valuable in helping to refine and strengthen the manuscript, and they have undoubtedly added clarity to the research. Please find the detailed responses below, along with the corresponding revisions and corrections in the re-submitted files. English was also revised and improved.

Comment 1: The introduction functions as a mini-review, repeating points on pathologization/medicalization and the history of intersex interventions. While the theoretical framing (Butler, Fausto-Sterling, Karkazis) is valuable, its length is disproportionate to the empirical section.

Response to Comment 1:

We thank the reviewer for this insightful observation. The introduction has been substantially revised to improve focus and proportionality. Historical details on pathologization and medicalization have been condensed (Line 58-136).

Comment 2: The paper invokes "sociocognitive factors" but doesn't manage to anchor them in a clear theoretical model. Concepts like attitudes, knowledge, and beliefs are treated descriptively rather than linked within a coherent framework. To strengthen the work, articulate and operationalize a specific theory that explicitly connects cognitive variables to attitudes and behaviors.

Response to Comment 2:

Thank you for the suggestion. The revised version now integrates a clearer theoretical foundation which provides an explicit framework linking knowledge, beliefs, and attitudes to behavioral intentions (Introduction).

Comment 3: The study relies on snowball and convenience sampling followed by post hoc "stratification", a methodologically inconsistent approach that undermines representativeness. Although selection bias is acknowledged, its potential impact is not quantified. The Spanish sample is then used to make cross-disciplinary claims, while the distribution across medical specialties remains unclear. To address these issues, specify recruitment channels and inclusion/exclusion criteria, clarify specialty representation, and justify the adequacy of the 210 participants for testing six hypotheses.

Response to Comment 3:

We appreciate the reviewer’s constructive feedback. The Procedure section (262-288) has been expanded to clarify recruitment channels, inclusion and exclusion criteria, and the rationale for the post hoc stratified sampling. Although the study employed an initial convenience and snowball approach, stratification was subsequently applied to ensure the representation of key professional subgroups (medicine, nursing, psychology, and social work). The distribution across medical specialties is now explicitly detailed in the Participants section (215-232). Selection bias and the limits of representativeness are further discussed in the Limitations section (678-683), where we acknowledge that the exploratory design and non-probabilistic sampling restrict the generalizability of findings. Nonetheless, the sample size (N = 210) was deemed adequate to test the six hypotheses using chi-square and ANOVA analyses, meeting the minimum cell count assumptions for these tests and providing sufficient statistical power for medium effect sizes.

Comment 4: The Intersex Knowledge Questionnaire is used without evidence of validation, and no psychometric properties are reported (e.g., factor structure, subscale reliability). "Attitudes" are operationalized with a small set of corrective vs. affirmative items that lack psychometric grounding. Category thresholds ("excellent", "adequate", etc.) appear to derive from mean SD, which can distort classification. Please report internal consistency (α, ω), conduct dimensionality analyses (EFA/CFA), and clarify whether any scales were adapted/translated and pretested.

Response to Comment 4:

We thank the reviewer for this valuable methodological observation. The Method section has been revised to explicitly state that the Intersex Knowledge Questionnaire (Bonavia & Palacios, 2020) has not yet undergone formal psychometric validation and was therefore employed as an exploratory instrument to capture preliminary dimensions of intersex-related knowledge and attitudes among healthcare professionals (257-259).

Category thresholds (e.g., excellent, adequate) were calculated using the sample’s empirical distribution and were applied consistently across dimensions for comparative interpretation. This clarification is now included both in the Method and Limitations sections.

Comment 5: The analysis addresses complex, multivariate questions, but reliance on chi-square tests and t-tests limits interpretability. Clarity would improve by using multivariable models (e.g., multiple regression or SEM) that adjust for key covariates such as age, profession, gender, and prior contact. Effect sizes and confidence intervals should be reported consistently, and the use of one-tailed p-values should be justified or avoided. These steps would yield a stronger, more nuanced picture of the results.

Response to Comment 5:

We sincerely thank the reviewer for this valuable methodological observation. We acknowledge that the use of chi-square tests, and t-tests offers a bivariate perspective on the relationships examined. These procedures were selected to identify preliminary patterns rather than to test a fully specified predictive model. This approach is consistent with early-stage empirical research designed to inform subsequent model development.

The manuscript has been revised to include effect sizes (Cramér’s V and Cohen’s d) and 95% confidence intervals across all applicable analyses, improving interpretability and transparency. A clarification has also been added to the Data Analysis section to explicitly state these reporting practices (329-331).

We agree that future research should apply multivariable models based on larger and more balanced samples, enabling the inclusion of relevant covariates such as age, profession, gender, and prior contact.

Comment 6: The crosssectional design does not support causal inference, yet several conclusions imply directional effects (e.g., "knowledge drives attitudes"). The discussion should also consider reverse causality, whereby professionals with more inclusive attitudes may be more likely to seek training. Rephrase claims to reflect correlational rather than causal relationships.

Response to Comment 6:

We appreciate the reviewer’s insightful observation. The Discussion (520-666) has been revised to explicitly acknowledge the potential for reverse causality, noting that professionals with more inclusive or affirmative attitudes may also be more inclined to pursue additional training or information about intersex healthcare.

Comment 7: Terminology is inconsistent: "intersexuality" and "intersex variations" are used interchangeably. Current practice favors intersex or variations of sex characteristics (VSC). Please standardize throughout and explain your usage at first mention. Ethical approval is reported, but the procedures for online recruitment - especially anonymity, data protection, and informed consent - need clearer description.

Response to Comment 7:

We sincerely thank the reviewer for this valuable and precise observation. The terminology has been thoroughly revised and standardized throughout the manuscript to ensure terminological consistency and alignment with contemporary ethical and academic standards. Additionally, the procedures for online recruitment have been clarified in the Methodology section, explicitly detailing the measures taken to ensure anonymity, data protection, and informed consent in accordance with ethical approval protocols (262-288).

Comment 8: The conclusions overstate the study’s implications given its methodological limitations. In particular, the claims that "Intersex healthcare attitudes are primarily driven by deficits 28 in training and the invisibility of intersex issues in medical education" (Abstract) and "These findings confirm that clinical attitudes are not explained solely by gender ste-730 reotypes but are primarily shaped by the availability of specialized training, institutional 731 organization, and the presence, or absence, of clear normative frameworks" are intriguing but not supported by sufficiently robust analyses. Please temper the language and align conclusions more closely with the descriptive evidence presented.

Response to Comment 8:

We thank the reviewer for this thoughtful observation. The language in the Conclusions and Discussion sections has been revised to moderate the claims and better align them with the descriptive nature of the analyses. Statements suggesting causal relationships have been rephrased to indicate possible associations.

Comment 9: Tables 3–9 are overly detailed and repetitive. Several could be moved to Supplementary Material.

We thank the reviewer for the suggestion. After careful consideration, tables were retained in the main text because each directly supports a specific hypothesis, aids result interpretation and preserve clarity and transparency. Given that the journal platform allows tables to be collapsed, we considered it appropriate to keep them accessible for readers who wish to consult the full data.

Comment 10: Reference list shows inconsistencies (missing DOIs, uneven citation formatting).

Response to Comment 10:

The entire reference list has been thoroughly reviewed and standardized to ensure consistency in formatting according to the journal’s citation style. Missing DOIs have been added whenever available.

Comment 11: Abstract should include sample size and main statistical results.

Response to Comment 11:

We thank the reviewer for this observation. The Abstract has been revised to include the information (22-31).

Comment 12: Some citations appear misformatted or missing reference details. See: Butler, J. El género en disputa: el feminismo y la subversión de la identidad; Paidós estudio; [1a ed., 12a reimpr.].; Paidós: Barcelona, 803 2020; ISBN 9788449320309.

Response to Comment 12:

Thank you. The entire reference list has been thoroughly reviewed and corrected. 

Reviewer 2 Report

Comments and Suggestions for Authors

Thank you for the opportunity to review your work.

STRENGTHS OF THE ARTICLE

a) The results and discussion section are presented very well.

b) The results are compelling.

c) The discussion has depth and objectivity.

Overall, the paper has great potential for catalyzing new ideas regarding how healthcare may be better managed for intersex individuals. 

Please see my suggestions below regarding some areas that need strengthening:

  1. The citations in the Introduction are more than 10 years old in some cases. It would greatly help the strength of this article if more recent (within 5 years) citations replaced these older citations/references. Thank you.
  2. There seems to be two types of citations styles, APA as well as a numeral system. Could the authors please clean this up? Thank you. 
  3. Lines 35-183 need to be reorganized and restructured. At present, it is a conglomeration of information, lacking cohesion to form a strong and precise academic argument for the research. Suggestion: Please relook the information contained in this section of the article and restructure this information to form a solid background focused on the sociocognitive factors associated with providing healthcare for intersex individuals.
  4. Also in the Introduction: where is the mention of gender, information and beliefs regarding intersex individuals? Are you focused on the gender, information and beliefs of intersex individuals or in the gender, information and beliefs of healthcare professionals? At present, the introduction is ambiguous and does not give clear direction of this. Please revise. Thank you. 
  5. It is only in 2.1 that the focus of the article is clear. With this in mind, please reframe the Introduction to meet the focus in 2.1. Thank you. 
  6. Please consider moving the information from lines 240-266 to be part of the results section. This is more results than methodology. In this section of the methodology the author(s) should focus on the inclusion and exclusion criteria for the participants, rather than their characteristics. 

Author Response

Thank you very much for taking the time to review this manuscript. We sincerely appreciate your thoughtful feedback and insightful suggestions. Your comments have been extremely valuable in helping to refine and strengthen the manuscript, and they have undoubtedly added clarity to the research. Please find the detailed responses below, along with the corresponding revisions and corrections in the re-submitted files.

Comment 1: The citations in the Introduction are more than 10 years old in some cases. It would greatly help the strength of this article if more recent (within 5 years) citations replaced these older citations/references. Thank you.

Response to Comment 1:

We thank the reviewer for this valuable observation. Additional recent references have been incorporated to strengthen the theoretical models supporting the study, while certain earlier works have been retained because the topic of intersex healthcare remains underexplored and foundational literature continues to provide essential conceptual grounding for current research.

Comment 2: There seems to be two types of citations styles, APA as well as a numeral system. Could the authors please clean this up? Thank you.

Response to Comment 2:

We appreciate the reviewer’s observation. The citation style has been carefully revised and standardized throughout the manuscript.

Comment 3: Lines 35-183 need to be reorganized and restructured. At present, it is a conglomeration of information, lacking cohesion to form a strong and precise academic argument for the research. Suggestion: Please relook the information contained in this section of the article and restructure this information to form a solid background focused on the socio-cognitive factors associated with providing healthcare for intersex individuals.

Response to Comment 3:

We thank the reviewer for this insightful observation. The introduction has been substantially revised to improve focus and proportionality. Historical details on pathologization and medicalization have been condensed. The revised version now integrates a clearer theoretical foundation which provides an explicit framework linking knowledge, beliefs, and attitudes to behavioral intentions (42-171).

Comment 4: Also in the Introduction: where is the mention of gender, information and beliefs regarding intersex individuals? Are you focused on the gender, information and beliefs of intersex individuals or in the gender, information and beliefs of healthcare professionals? At present, the introduction is ambiguous and does not give clear direction of this. Please revise. Thank you.

Response to Comment 4:

We thank the reviewer for this insightful observation. The introduction has been revised to clarify the study’s focus on the gender beliefs, information, and attitudes of healthcare professionals toward intersex people, rather than on the experiences or beliefs of intersex individuals themselves. Additionally, the section now integrates a clearer theoretical foundation linking knowledge, beliefs, and attitudes to behavioral intentions, ensuring coherence with the socio-cognitive model guiding the research (42-171).

Comment 5: It is only in 2.1 that the focus of the article is clear. With this in mind, please reframe the Introduction to meet the focus in 2.1. Thank you.

Response to Comment 5:

Thank you. Changes have been made (42-171).

Comment 6: Please consider moving the information from lines 240-266 to be part of the results section. This is more results than methodology. In this section of the methodology the author(s) should focus on the inclusion and exclusion criteria for the participants, rather than their characteristics.

Response to Comment 6:

We thank the reviewer for this helpful observation. The extra information previously included in the Methodology section has been relocated to the Results section (333-351). In addition, the inclusion and exclusion criteria have been made more explicit in the Methodology section to improve clarity and transparency (218-221).

Reviewer 3 Report

Comments and Suggestions for Authors

Many thanks to the authors for their text.
The study is interesting and covers an underexplored topic.

I share some specific comments:

• The abstract could slightly summarize the introductory section (the first three lines are very contextual) to gain more space in the results. It should also include the sample size directly in the first methodological sentence.
• Although the critical tone of the introduction is appropriate, it could be balanced with brief references to institutional progress (e.g., the gradual incorporation of clinical guidelines and bioethical reforms in Europe). Furthermore, some paragraphs are dense and could be divided to improve readability.
• Regarding methodology, the Intersex Knowledge Questionnaire lacks formal psychometric validation; it would be helpful to acknowledge this not only in the Limitations section, but also in the Method section, noting that this is an exploratory instrument.
• It would be desirable to specify the inclusion/exclusion criteria more precisely-
• Briefly justify the decision to combine chi-square and ANOVA analyses when the dependent variables are of a different nature.
• It would be helpful to add a brief interpretative paragraph at the end of each subsection of the results to reinforce the connection with the hypotheses.
• The discussion is extensive. It could benefit from a final summary by subsection.
• The conclusion requires a brief projection of future lines of research.

Author Response

Thank you very much for taking the time to review this manuscript. We sincerely appreciate your thoughtful feedback and insightful suggestions. Your comments have been extremely valuable in helping to refine and strengthen the manuscript, and they have undoubtedly added clarity to the research. Please find the detailed responses below, along with the corresponding revisions and corrections in the re-submitted files.

Comment 1: The abstract could slightly summarize the introductory section (the first three lines are very contextual) to gain more space in the results. It should also include the sample size directly in the first methodological sentence.

Response to Comment 1:

We appreciate the reviewer’s observation. The introductory section of the abstract has been condensed to reduce contextual background and emphasize the study’s aims and key findings. Additionally, the sample size (210 healthcare professionals) has been explicitly included at the beginning of the Methods sentence, as suggested (12-22).

Comment 2: Although the critical tone of the introduction is appropriate, it could be balanced with brief references to institutional progress (e.g., the gradual incorporation of clinical guidelines and bioethical reforms in Europe). Furthermore, some paragraphs are dense and could be divided to improve readability.

Response to Comment 2:

We appreciate the reviewer’s constructive suggestion. We agree that the critical tone of the introduction benefits from being balanced with references to recent institutional and regulatory progress in Europe. Accordingly, we have added that information (114-136).

Comment 3: Regarding methodology, the Intersex Knowledge Questionnaire lacks formal psychometric validation; it would be helpful to acknowledge this not only in the Limitations section, but also in the Method section, noting that this is an exploratory instrument.

Response to Comment 3:

We appreciate this observation. The manuscript has been revised accordingly. The Method section now explicitly notes that the Intersex Knowledge Questionnaire (Bonavia & Palacios, 2020) has not yet undergone formal psychometric validation and is therefore used as an exploratory instrument to capture preliminary dimensions of intersex-related knowledge and attitudes among healthcare professionals (278-259). This clarification complements the statement already included in the Limitations section. (line 678-682), and conclusions (739-745).

Comment 4: It would be desirable to specify the inclusion/exclusion criteria more precisely.

Response to Comment 4:

We appreciate the reviewer’s comment. The Method section now clearly specifies the inclusion and exclusion criteria (218-221).

Comment 5: Briefly justify the decision to combine chi-square and ANOVA analyses when the dependent variables are of a different nature.

Response to Comment 5:

Thank you for the comment. The combination of parametric (t-tests, ANOVA) and non-parametric (chi-square) analyses was based on the measurement level of the variables. Chi-square tests were used for categorical data, while parametric tests were applied to continuous scores.

Comment 6: It would be helpful to add a brief interpretative paragraph at the end of each subsection of the results to reinforce the connection with the hypotheses.

Response to Comment 6:

Thank you for the suggestion. Brief interpretative paragraphs have been added at the end of each Results subsection to reinforce the connection between the findings and the corresponding hypotheses (409-411; 430-432; 450-451; 466-468; 482-485; 487-489; 513-516)

Comment 7: The discussion is extensive. It could benefit from a final summary by subsection.

Response to Comment 7:

We appreciate the reviewer’s suggestion. The Discussion section has been revised to improve clarity and coherence. Rather than adding new summary paragraphs, which could lead to repetition, the section was streamlined and reorganized to make the main findings and their implications easier to follow (520-666).

Comment 8: The conclusion requires a brief projection of future lines of research.

Response to Comment 8:

The conclusion section has been revised to include a brief projection of future research lines (731-745).